# Effects of Catalyst Pretreatment on Carbon Nanotube Synthesis from Methane Using Thin Stainless-Steel Foil as Catalyst by Chemical Vapor Deposition Method

**DOI:** 10.3390/nano11010050

**Published:** 2020-12-28

**Authors:** Thuan Minh Huynh, Sura Nguyen, Ngan Thi Kim Nguyen, Huan Manh Nguyen, Noa Uy Pham Do, Danh Cong Nguyen, Luong Huu Nguyen, Cattien V. Nguyen

**Affiliations:** 1Petrovietnam R&D Center for Petroleum Processing, Vietnam Petroleum Institute, Lot E2b-5, D1 Street, Saigon Hi-Tech Park, District 9, Hochiminh City 708400, Vietnam; suran.pvpro@vpi.pvn.vn (S.N.); nganntk@vpi.pvn.vn (N.T.K.N.); huannm.pvpro@vpi.pvn.vn (H.M.N.); uydpn@vpi.pvn.vn (N.U.P.D.); luongnh.pvpro@vpi.pvn.vn (L.H.N.); 2Saigon Hi-Tech Park Research Laboratories, Lot I3, N2 Street, Saigon Hi Tech Park, District 9, Hochiminh City 708400, Vietnam; danh.nguyencong@shtplabs.org; 3NTherma Corporation, 458 S Hillview Dr, Milpitas, CA 95035, USA; cattien.nguyen@ntherma.com

**Keywords:** carbon nanotubes, stainless-steel foil catalyst, chemical vapor deposition, semiconductor, pretreatment

## Abstract

Synthesis of carbon nanotubes (CNTs) was carried out using methane as a carbon source via the chemical vapor deposition (CVD) method. A thin stainless-steel foil was used as catalyst for CNT growth. Our results revealed that pretreatment step of the stainless-steel foil as a catalyst plays an important role in CNT formation. In our experiments, a catalyst pretreatment temperature of 850 °C or 950 °C was found to facilitate the creation of Fe- and Cr-rich particles are active sites on the foil surface, leading to CNT formation. It is noted that the size of metallic particles after pretreatment is closely related to the diameter of the synthesized CNTs. It is interesting that a shorter catalyst pretreatment brings the growth of semiconducting typed CNTs while a longer pretreatment creates metallic CNTs. This finding might lead to a process for improving the quality of CNTs grown on steel foil as catalyst.

## 1. Introduction

Carbon nanotubes (CNTs) and graphene are the most promising materials for various applications due to their outstanding characteristics such as high thermal and electrical conductivities, optical absorption, and mechanical strength. With a cylindrical structure and nanometer-sized diameter, CNTs have attracted more attention from academic and industrial scientists due to their promising market sizes and outstanding applications e.g., in batteries [1,2], in lubricant additives [3,4,5], and in atomic force microscopy (AFM) [6,7,8]. Currently, the most popular method for CNT synthesis is via hydrocarbon decomposition at high temperature in the presence of a catalyst. Chemical vapor deposition (CVD) is considered an advanced method for CNT production on industrial scale due to high yield and high purity of the product obtained from a low growth temperature (less than 1000 °C) and simple furnace [9,10]. Hydrocarbons are the preferred feedstock for CNT synthesis using the CVD method with the catalyst in a powder form or as a thin film on a substrate surface. A powder catalyst is now preferred in commercial production due to its easy fabrication. However, a considerable amount of waste is produced, and its treatment is compulsory [11,12,13]. The cost for waste treatment and purification steps are very high and contribute significantly to the total CNT production cost. On the other hand, a thin-film catalyst can produce higher-quality CNTs without purification steps.

The yield of CNTs from methane is definitely lower than those from other carbon sources, such as ethylene, propylene, acetylene, benzene, methanol, ethanol and so on. Generally, the temperature for methane decomposition is significantly higher than those for other carbon sources [14]. Transition metals such as Ni, Fe, and Co are widely used as catalysts for methane decomposition to produce CNTs [15,16,17]. Avdeeva performed the decomposition of methane with Fe/Al_2_O_3_, Co/Al_2_O_3_ and Ni/Al_2_O_3_ catalysts and the conversion rates were 2%, 7%, and 15%, respectively [18]. Muradov reported that conversion of methane was 28% at 950 °C with an activated carbon as the catalyst [19]. Yahyazadeh and B. Khoshandam synthesized CNTs from methane via the CVD method in the presence of iron, molybdenum, and iron-molybdenum alloy thin-layer catalysts. It has been reported that the main solid product is multiwalled carbon nanotube (MWCNT) with its diameter in the range of 16–55 nm. The optimal condition for CNT synthesis is 750 °C, 1 atm and an iron-based catalyst supported on quartz substrate [20]. In addition, X. Lepro et al. reported the growth of yarn-spinnable and sheet-drawable carbon nanotube forests on highly flexible stainless-steel sheets, instead of the conventionally used silicon wafers. The authors used Fe deposited on stainless-steel layer as a catalyst at the typical conditions (feedstock: acetylene or ethylene, 700–760 °C, 1 atm) [21]. It is interesting that a research group from Institute of Materials Science, Vietnamese Academy of Science and Technology investigated CNT synthesis from acetylene using CT3 steel foils as catalysts [22]. It has been found that carbon nanotubes were formed on steel foils at 600–900 °C in 30 min, and the optimal condition for CNT synthesis was 800 °C. In addition, the catalysts can be reused a few times without any treatment. However, in this study, changes in catalyst topology and properties with time on stream have been not reported. Up to now, in the literature, we have not found any work related to CNT synthesis from methane as feedstock using stainless-steel foil as a catalyst. In this study, CNTs were successfully synthesized from methane using a thin stainless-steel foil as a catalyst via the CVD method. The effects of catalyst pretreatment temperature and activation time onto CNT growth are discussed and the mechanism of active sites formation on the stainless-steel foil for CNT growth is proposed.

## 2. Materials and Methods

### 2.1. Stainless-Steel Foil

In this study, a commercial stainless-steel foil with a thickness of approximately 40 μm, provided by NTherma Corporation (Milpitas, CA 95035, USA), was used as a catalyst precursor. It was cut to a size of 1 × 1 inch and used directly without any treatment. The SEM-EDX (Scanning Electron Microscopy–Energy-dispersive X-ray Spectroscopy) in Table 1 reveals that the main metallic components of the catalyst are Fe, Cr, and Al. The EDX spectra are depicted in Appendix A.

### 2.2. Catalyst Pretreatment and CNT Synthesis

The catalyst pretreatment and CNT synthesis were conducted in fully automated quartz-tube horizontal-type thermal CVD equipment (CN-CVD-200TH, ULVAC, Inc., Kanagawa, Japan). The experimental setup of the CVD is depicted in Figure 1. In general, a catalyst foil was placed in the middle of a quartz tube installed in an infrared heating furnace (ULVAC, Inc., Kanagawa, Japan). The steel foil with a size of 1 × 1 inch was inserted into the CVD quartz reactor with a high temperature furnace and mass flow controllers. Normally, argon (Messer, 99.99% pure) flow of 1000 standard cubic centimeters per minute (sccm) is used to purge the reactor.

For catalyst pretreatment, the furnace was heated in an argon flow of 1000 sccm to the given pretreatment temperature (e.g., 750 °C, 850 °C, and 950 °C) and was maintained at a programmed time (e.g., 10 min or 2 min). The furnace was then cooled down in argon gas to room temperature. The catalyst was taken out and used for analysis.

For CNT synthesis, the pretreated catalyst was used for CNT growth at the same temperature. At this temperature, the methane gas (Messer, 99.99% pure) with a flow of 100 sccm was blown into the furnace for a programmed time (30 min) for CNTs to be grown. After reaction, the methane gas was switched off and the furnace was then cooled down to room temperature. CNTs grown on the foil were collected and characterized.

### 2.3. Catalyst and Product Characterization

The catalyst and the formed CNTs were used for characterization and described as follows.

The crystal phase of the obtained CNTs was determined using X-ray diffraction (XRD) (Bruker D2 Phaser with λCuKα = 1.54 Ȧ radiation at 30 kV and 10 mA. The elemental composition of CNTs was obtained from the energy-dispersive X-ray spectrum (EDX) (H-7593 Horiba, Kyoto, Japan) at 15 kV. The morphology of the synthesized CNTs was observed using scanning electron microscopy (SEM) (FE SEM S4800 Hitachi, Tokyo, Japan). The operating voltage and current were 10 kV and 10 μA, respectively. The structure of CNTs was also determined by using high resolution transmission electron microscopy (HR-TEM) (JEOL JEM-2100) at 200 kV with a magnification of 800,000. Raman spectroscopy was used to confirm the growth and determine the structural purity of the synthesized CNTs (HORIBA XploraOne 532 nm, Kyoto, Japan).

In addition, thermogravimetric analysis (TGA) was used to analyze the purity of the CNTs (SETARAM LabSys Evo STA, Caluire, France) under flowing air at temperatures ranging from 100 to 1000 °C with a heating rate of 10 K/min.

## 3. Results and Discussion

### 3.1. Effects of Catalyst Pretreatment Temperature on CNT Growth

The three values of catalyst pretreatment temperature were investigated, namely 750 °C, 850 °C, and 950 °C. The SEM images in Figure 2 show that there was no CNT formation at the temperature of 750 °C (see Figure 2a). This finding is consistent with our Raman results in the following section. However, at a higher temperature (at 850 °C at 10 min pretreatment, see Figure 2b), the CNTs were formed and their diameter was in the range of 20–80 nm.

The Raman analysis shows in Figure 3 reveals that CNTs are formed at the pretreatment temperature of 850 °C and 950 °C, whereas no evidence for CNT formation is observed at 750 °C treatment. For synthesized CNTs, their Raman spectra show several important peaks, e.g., D band (~1355 cm^−1^), G^−^ band (~1575 cm^−1^), and G^+^ band (~1650 cm^−1^) and G′ band (~2620 cm^−1^). The spectrum shows the formation of two peaks G and G′ at 1575 cm^−1^ and 2620 cm^−1^, which are related to the sp^2^ carbon form in the graphene structure [23,24]. It is interesting to find that CNTs obtained at 850 °C possess G^−^ and G^+^ bands in their Raman spectrum, as the result of presence of single-walled carbon nanotubes (SWCNTs). On the other hand, both SWCNTs and MWCNTs have been also observed in their TEM images (see Figure 12). According to Kumar and Ando [25], SWCNTs might be formed over infinitesimal metallic nanoparticles as catalyst under low temperature (<900 °C). The formation of CNT mixture was also observed in [26]. In addition, for CNTs obtained at 850 °C, their I_D_/I_G_ ratio of 0.54 (<1) indicates a good degree of graphitization. In fact, it is higher than the normal I_D_/I_G_ ratio of SWCNTs and lower than that of MWCNTs as shown in reference [27]. However, as for CNTs synthesized at 950 °C, only G band has been observed in their Raman spectrum, attributed to MWCNTs. Generally, metallic nanoparticles are subject to be sintered under high temperature, resulting in the formation of bigger particles (as mentioned below in AFM analysis, see Figure 8) leading to priority of MWCNT formation. It is worth noting that our Raman analysis also shows the influence of the pretreatment temperature on the types of the as-synthesized CNTs. In terms of CNTs formed at 850 °C of pretreatment, the G^−^ band indicates the Lorentzian line shape corresponding to semiconducting CNTs. In contrast, the G^−^ band of CNTs formed at 950 °C is overlapped with the G^+^ band as mentioned above, and it is normally deconvoluted by a Breit–Wigner–Fano line which is attributed to metallic type [28,29].

The morphology and structure of the catalyst precursor was first investigated using SEM and EDX methods. The SEM image of the fresh catalyst shows that grain boundary phase is observed (see Figure 4).

Figure 5 displays SEM images of the catalyst precursor and its postactivation. Interestingly, the formation of metallic nanosized particles on the catalyst precursor surface after a thermal pretreatment can be observed. The mechanism for the formation of those species has not been well understood yet. It can be presumed that the pretreatment step results in transformation of metallic phases in the thin foil into a number of particles on its surface and they can serve as active sites for CNT growth. The SEM images in Figure 5 also reveals that the particles are distributed across the entire surface, especially at 850 °C pretreatment. It can be seen that the temperature should be an important parameter affecting the density and size of particles on the catalyst precursor surface. It is worth noting that the size of metallic species is smaller as the pretreatment temperature is increased. This finding is consistent with the results of Rosales et al. [30].

It should be mentioned that alpha-iron and chromium oxide are found significantly in the XRD pattern of the catalyst precursor. After pretreatment at 850 °C, the formation of chromium carbide is observed and the peak of alpha-iron is slightly shifted to the right (see Figure 6) [31].

However, a pretreatment at very high temperature may be leading to the melting issue due to the Ostwald ripening phenomenon and, as a result, small metal particles were destroyed. Figure 7 shows a certain number of melting positions in the SEM image of the catalyst after pretreatment at 950 °C.

To deeply understand the catalyst surface morphology, the composition of metallic particles on its surface after pretreatment was analyzed and the result is described in Table 2. The EDX spectra are depicted in Appendix A. As can be seen, the compositions of Fe and Cr in the metal particles after pretreatment at 850 °C or 950 °C are slightly higher compared to those at 750 °C. The result is consistent with Han et al. [32] who treated the catalyst surface with composition (Fe~66%wt.; Al~6%wt. and Cr~18%wt.). The result shows that Cr phase was increased and found on the surface after treatment at suitable temperature and time. This result also supports the finding from the above-mentioned XRD patterns.

Therefore, the results from SEM and SEM-EDX reveal that the formation with high density of Fe- and Cr-rich particles has been observed on the catalyst surface after its pretreatment at 850 °C and 950 °C. Interestingly, the Fe and Cr are well known as catalysts and/or promoters for methane decomposition to produce CNTs [33,34].

The morphology of the pretreated catalyst was also analyzed by using the AFM method. The AFM images are given in Figure 8. It can be clear that the nanoparticles are well-distributed over its surface after 850 °C pretreatment. It is also observed that the melting process occurs at the pretreatment of 950 °C, whereas the particles are not well-formed at 750 °C. The distribution of particles is given in Figure 9. It can be seen that the particles are in the range of 10–25 nm and well-established at 850 °C. At the 750 °C or 950 °C pretreatment, a wide range of particle sizes was observed. The result is in line with the above-mentioned SEM result.

### 3.2. Effects of Catalyst Pretreatment Time on CNT Growth

As we know, the structural features of carbon nanotubes grown by the CVD method depend on the experimental conditions (e.g., the catalyst pretreatment and synthesis conditions). The catalyst pretreatment time also affects the density and diameter of CNTs on the catalyst. In our study, at 850 °C, pretreatment was carried out over two durations, namely two and 10 minutes. With two minutes’ pretreatment, the CNTs’ density is higher, their diameter is in the range of 10–20 nm, and their length is app. 300 nm (see Figure 10). It is interesting to note that the size of metallic particles is closely related to the diameter of the synthesized CNTs. Therefore, a good control of metallic particle formation on the catalyst surface can produce high-quality CNTs.

The Raman spectra of synthesized CNTs obtained from different pretreatment times are depicted in Figure 11. It can be seen that a shorter pretreatment time brings CNTs with a high intensity of the G^+^ band, whereas the G^−^ band shows the reverse direction followed by decreasing the semiconducting CNTs. The I_D_/I_G_ ratio of CNTs formed at 850 °C pretreatment for 2 min and 10 min are 0.53 and 0.54, respectively. Obviously, the pretreatment time of the catalyst precursor also plays a significant role on the type but not on the quality of the CNTs formed.

In order to identify the number of walls of the synthesized CNTs, HR-TEM was used. The TEM images shown in Figure 12 reveal that the CNTs formed at 850 °C possessed various types, including single-walled (bundles of single-walled CNTs, which is similar to the finding in [35]), double-walled, and multiwalled carbon nanotubes (six walls). This observation is consistent with our Raman results.

To quantify the purity of the formed CNTs, TGA analysis was used. TGA results of the synthesized CNTs reveal that they possess almost 100% of nanocarbons and the amorphous phase was not found (see Figure 13).

### 3.3. Proposing the Mechanism of Active Sites Formation on the Stainless-Steel Foil for CNT Growth

Stainless-steel foil contains metallic components (e.g., iron and chromium) that can be used as a catalyst for CNT synthesis (see Table 1). However, these components are “inactive” due to their free-form existence in the foil. It is necessary to transform metallic phases in the foil into metallic active sites. In our study, it has been observed that a thermal pretreatment is an effective method to bring positive influences on the formation of active sites on the steel-foil surface. In fact, under the pretreatment at 850–950 °C, we can observe that the formation of metallic particles on the catalyst precursor surface facilitates CNT growth. It seems to be that a suitable temperature results in a rearrangement of metallic phases in the foil, leading to formation of iron- and chromium-rich particles on its surface (see Table 2). They serve as active sites for CNT growth from methane decomposition. As a result, as soon as the gas hits the pretreated foil surface, they are decomposed in contact with the formed particles and CNTs are produced (Figure 14). The foil pretreatment condition has been observed to play a significant role in CNT growth. Therefore, further investigation should be carried out for a good control of active site formation.

## 4. Conclusions

In this study, we investigated the CNT synthesis based on a commercial stainless-steel foil as a catalyst using methane as a carbon source precursor. Single-walled, double-walled, and multiwalled carbon nanotubes (six walls) were grown at 850 °C while only multiwalled carbon nanotubes were produced at 950 °C. Proper pretreatment of the stainless-steel foil is crucial for the successful synthesis of CNTs using the CVD method. Under the pretreatment temperature of 850–950 °C, iron- and chromium-rich particles are formed on the steel foil surface and serve as active sites for CNT growth. The metal support appears to increase the surface roughness and provide more active nucleation sites for CNTs formation. Thus, further investigations should be conducted to bring the active site formation under control. In addition, the pretreatment time plays a significant role in the type of CNTs formed.

In contrast to other methods, the present catalyst could easily be applied to large thin foil surfaces and geometries and has no scale up limitation. The technique is inexpensive and capable of producing a uniform layer of CNTs on stainless steel. The CO_2_ cofeeding and optimization process are under investigation and will be submitted in another manuscript.

## Figures and Tables

**Figure 1 nanomaterials-11-00050-f001:**
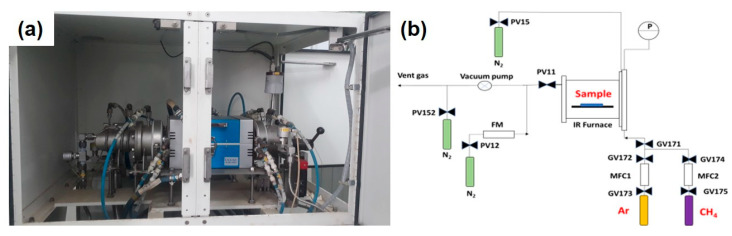
Experimental CVD set-up: (**a**) digital photo; (**b**) schematic diagram.

**Figure 2 nanomaterials-11-00050-f002:**
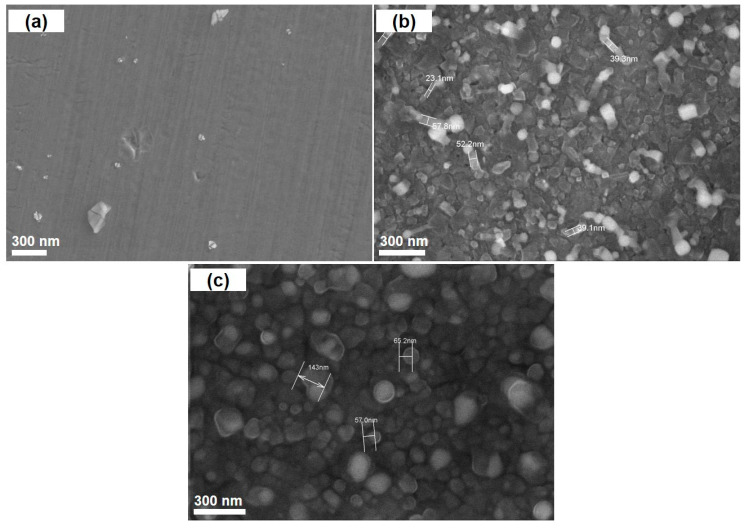
SEM images of the synthesized carbon nanotubes (CNTs) at different pretreatment temperatures: (**a**) at 750 °C for 10 min; (**b**) at 850 °C for 10 min; (**c**) at 950 °C for 10 min.

**Figure 3 nanomaterials-11-00050-f003:**
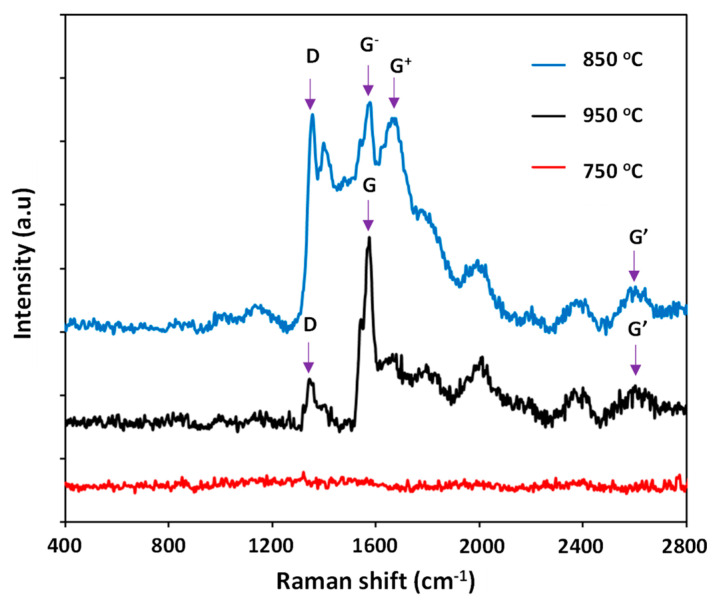
Raman spectra of the formed CNTs at different pretreatment temperatures.

**Figure 4 nanomaterials-11-00050-f004:**
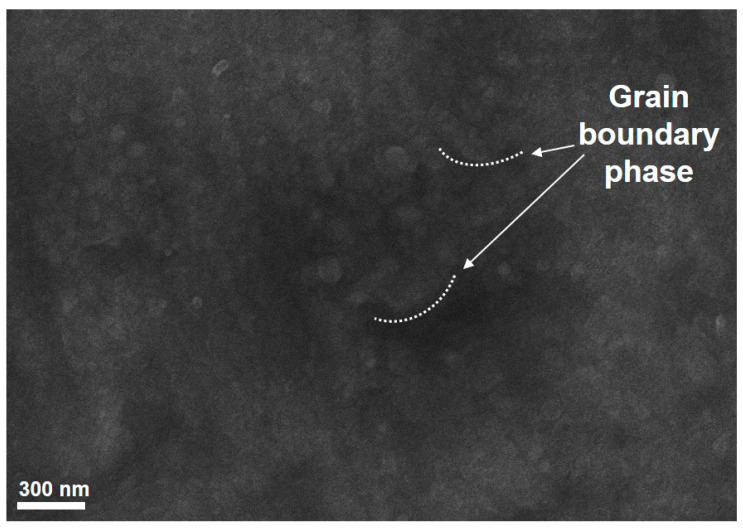
SEM images of the catalyst precursor.

**Figure 5 nanomaterials-11-00050-f005:**
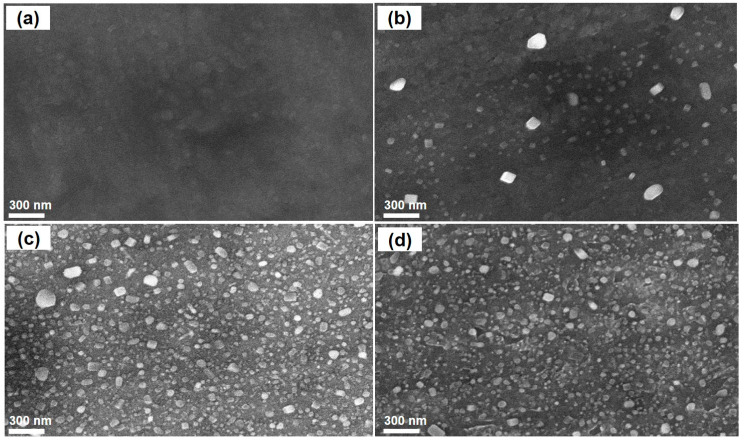
SEM images of the stainless-steel foil (**a**) and after pretreatment at different temperatures for 10 min: (**b**) 750 °C; (**c**) 850 °C; (**d**) 950 °C.

**Figure 6 nanomaterials-11-00050-f006:**
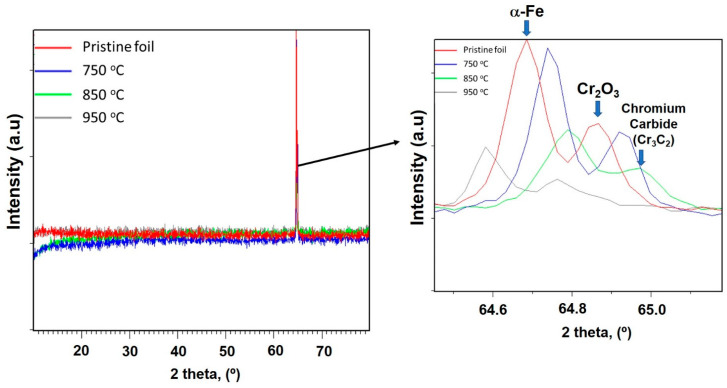
XRD patterns of the fresh and after pretreatment catalyst.

**Figure 7 nanomaterials-11-00050-f007:**
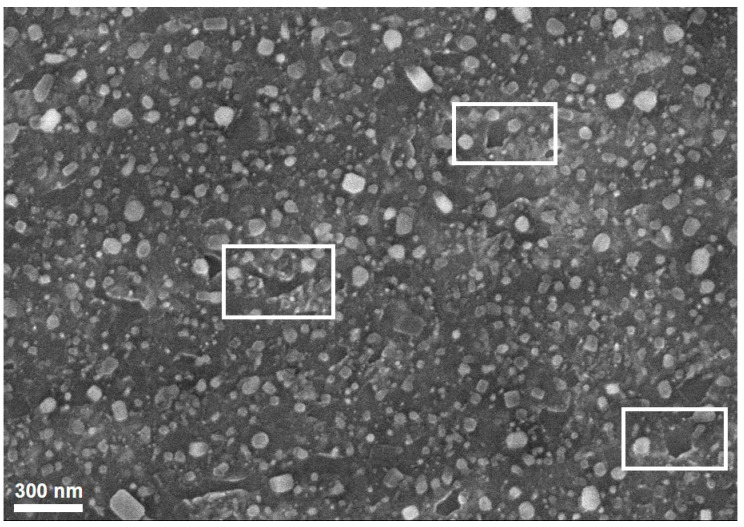
Melting points in the SEM image of the catalyst after pretreatment at 950 °C. Rectangles indicate melting point area.

**Figure 8 nanomaterials-11-00050-f008:**
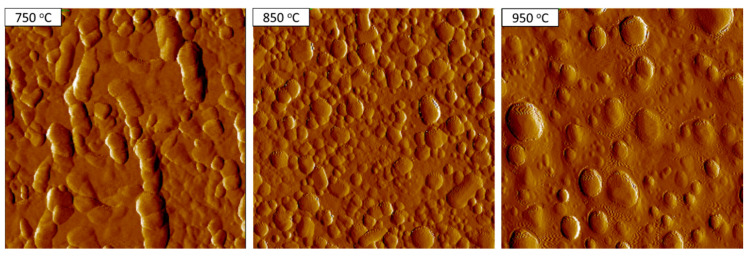
Atomic force microscopy (AFM) images of the catalysts after pretreatment at different temperatures for 10 min.

**Figure 9 nanomaterials-11-00050-f009:**
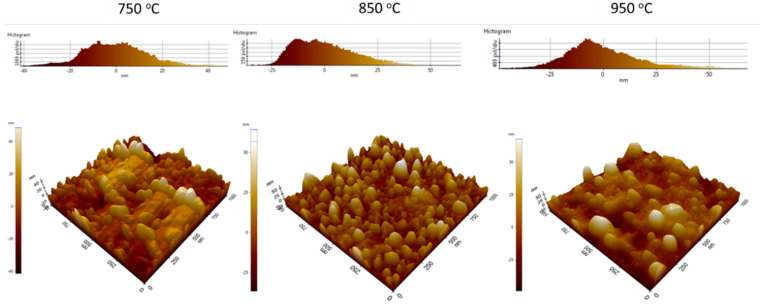
Distribution of particles on surface using AFM method.

**Figure 10 nanomaterials-11-00050-f010:**
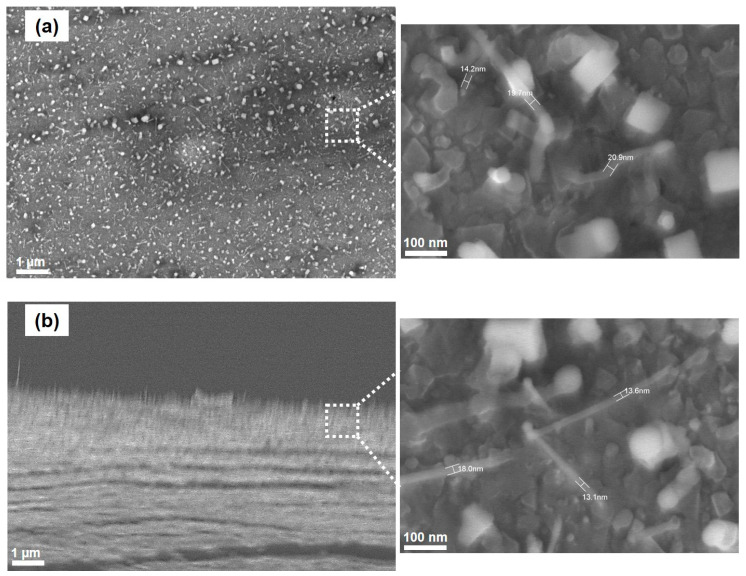
SEM images of formed CNTs at 850 °C for 2 min: (**a**) vertical direction and (**b**) horizontal direction.

**Figure 11 nanomaterials-11-00050-f011:**
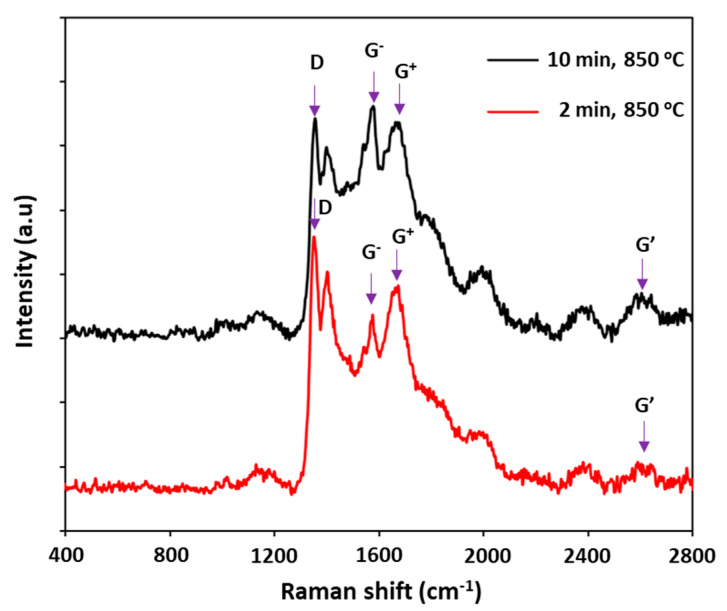
Raman spectra of the formed CNTs at different pretreatment times.

**Figure 12 nanomaterials-11-00050-f012:**
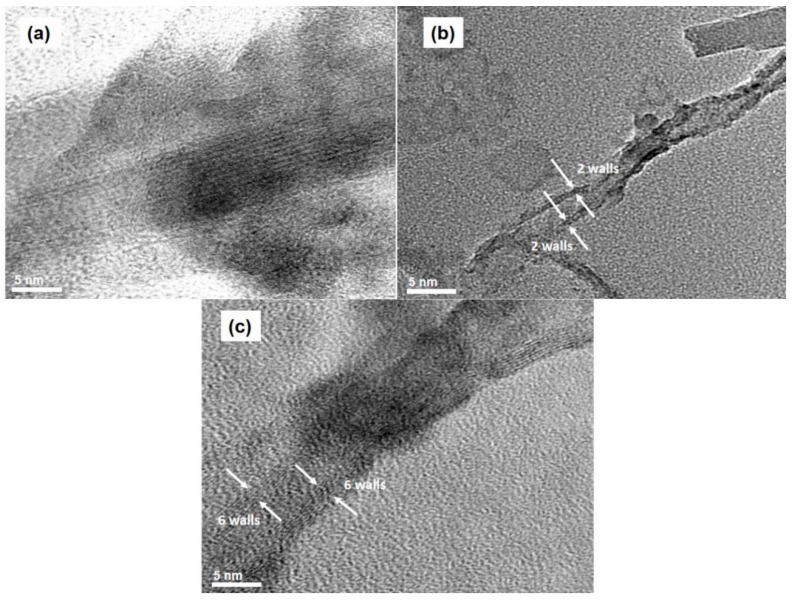
TEM images of synthesized CNTs at the temperature of 850 °C: (**a**) bundles of single-walled, (**b**) double-walled, and (**c**) multiwalled CNTs.

**Figure 13 nanomaterials-11-00050-f013:**
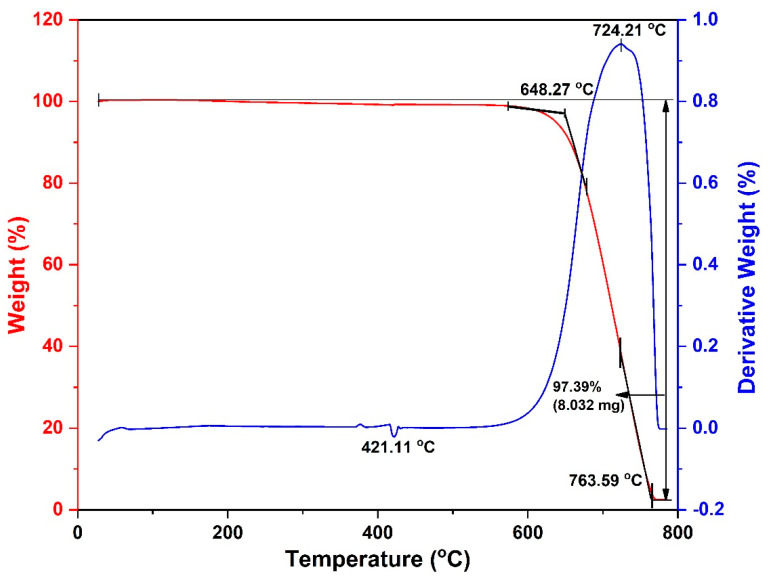
TGA curves of synthesized CNTs at the temperature of 850 °C.

**Figure 14 nanomaterials-11-00050-f014:**
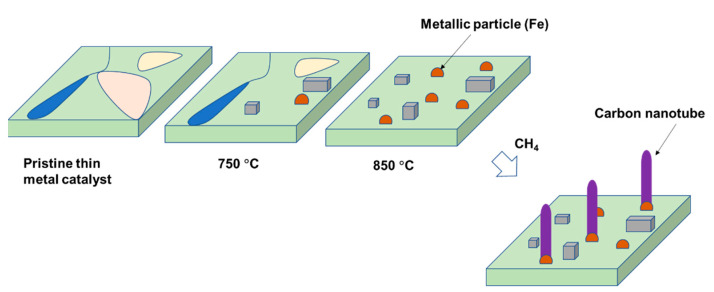
Schematic representation of the mechanism of CNT growth over thin-foil steel catalyst.

**Table 1 nanomaterials-11-00050-t001:** Composition of the catalyst precursor based on energy-dispersive X-ray spectrum (EDX).

Element	Composition, % Atom
C	8.5
O	16.2
Al	9.7
Cr	15.1
Fe	50.5

**Table 2 nanomaterials-11-00050-t002:** Composition of the metallic particles on the surface of the catalyst precursor after pretreatment.

Elements	Atom.% after Pretreatment at Temperature of			Atom.% of Stainless-Steel Foil (From Table 1)
	750 °C	850 °C	950 °C	
C	9.9	6.1	8.3	8.5
O	11.8	10.6	11	16.2
Al	10.9	11.1	11.9	9.7
Cr	15.2	17.3	16.0	15.1
Fe	52.2	54.9	52.8	50.5

## Data Availability

Data available in a publicly accessible repository.

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
