# Peer review of "Effects of Catalyst Pretreatment on Carbon Nanotube Synthesis from Methane Using Thin Stainless-Steel Foil as Catalyst by Chemical Vapor Deposition Method"

_nanomaterials, 2020, doi:10.3390/nano11010050_

Round 1

Reviewer 1 Report

Compared to the previous version the manuscript is much much better.

I recommend its publication after only few English corrections indicated in the attached manuscript

Author Response

Thank you for your comment. We have corrected accordingly some English points which are indicated in your attached file (see below). All changes have been highlighted in yellow in the revised manuscript. 

- Line 47: The word “Transitional” has been turned into “Transition”.

- Line 64: The word “for” has been changed into “related to”.

- Line 248: The word “investigate” has been corrected as “investigated”.

Reviewer 2 Report

Revised manuscript is much improved and is acceptabe.

Author Response

Thank you for your support.

Reviewer 3 Report

The authors addressed major concerns from previous reviewers, and incorporate major changes that enhanced the quality of the present manuscript.

This reviewer, may agree or disagree with other opinions. However, the technical sound seems suitable to publish in nanomaterial. 

This reviewer supports the publication of this version

Author Response

Thank you for your support.

This manuscript is a resubmission of an earlier submission. The following is a list of the peer review reports and author responses from that submission.

Round 1

Reviewer 1 Report

In this work, Huynh et al. studied the effect of catalyst pre-treatment on CNT synthesis from methane using a thin stainless-steel foil as catalyst by CVD method. The overall results are very poor. The samples show too much impurities and are characterized using limited number of techniques. It is hard to make good conclusion from the results (including all Figures) shown in this paper. The manuscript is not well presented with many of the scientific discussions are just hypothesis. Therefore this paper must not be accepted for publication in Nanomaterials.

Some specific comments that may improve the quality of this paper are:

- Introduction is too short and lack of information. In addition, only 10 references have been used in Introduction. Some sentences that require citations do not have any reference.

- The authors must provide every single details in Materials and Methods part. For example, what type of stainless-steel foil has been used? where is the source?

- Table shows the elemental composition from the SEM-EDX (as authors stated). The authors should provide the EDX spectrum of this result.

- The authors should improve the quality of Figure 1. Also, it is better to label the Figures as Figure 1a and 1b with 1a being digital photo and 1b is schematic diagram.

Author Response

Thanks for your valuable comments. Please see the attachment for our response. 

Reviewer 2 Report

Effects of catalyst pre-treatment on carbon nanotube synthesis from methane using thin stainless-steel foil as catalyst by chemical vapor deposition method

By Thuan Minh Huynh and all

The paper describes new handy material which upon thermal pre-treatment is suitable as a catalyst for synthesis of carbon nanotubes using methane as carbon source by CVD technique. Dependent on the time and temperature of the pre-treatment the catalyst to some extent favors formation of metallic and semiconducting MWCNTs. The structure of the catalyst surface is fully characterized; the kind of product is correlated with the reaction conditions and properties of the catalyst surface.

Text and Figures are well ordered and presented, but there are also many parts of the manuscript that require significant corrections. All are listed below, some with suggested changes or discussions. They are also highlighted in the original manuscript. While the manuscript is good, linguistic errors and ambiguities do not qualify it for printing as it is.

Line:

32-33: CNTs are attracted more attention from academic and industrial scientists due to their promising market sizes and outstanding applications – note some industrial applications e.g. in batteries or in AFM.

51-52: catalyst fabricated by Fe SUPPORTED on stainless steel layer as substrate consider DEPOSITED

54-55: CNT was fordem consider: carbon nanotube was fordem Or carbon nanotubes were formed

55: The optimal condition for CNT synthesis is 800 °C and during 30 minutes on stream. Correct this sentence

65-66: In this study, a commercial stainless-steel foil with a thickness of approximately 40 um

WAS used as a catalyst, WAS provided by NTherma corporation Consider correcting this sentence

66: 1 in? does it mean 1 inch?

67: compositions – consider components

73: in middle – in the middle?

73: tube in the status of face up – the tube with open top?

75: reactor which consists of a quartz tube -  could be replaced by quartz reactor

77: 1000 – 1,000

83: was  -> were

84: Figure 1. Experimental set-up of the CVD - Figure 1. Experimental CVD set-up

88: determined times – established? Programmed, coded? times

94 and 98: 15kV -> 15 kV; 200kV  -> 200 kV

98: 800,000 x – what does it mean? Magnification?

107: in consistent what does it mean: inconsistent or consistent???

108: (at 850 °C at 10 min pretreatment, see Figure 8b), à pretreated for 10 min at 850 °C. the reference to Fig 8b makes no sense!

113-120: this paragraph should be very carefully corrected in better English

127 Figure 4. SEM images of the fresh catalyst  -- is it possible to enhance a little bit the contrast?

130-136: This paragraph should be corrected using proper English

138: Figure 5. SEM images of the fresh catalyst and after pretreatment Could the phrase fresh catalyst  be replaced by the stainless steel foil???

144: Figure 6. XRD patterns of the fresh and after pretreatment catalyst. – increase the inset showing temperature.  What about formation of iron carbide?

145: of – at?

151-159: This paragraph should be in good English

160: are really formed metal particles?

172: different pretreatment temperatures in 10 min – after pretreatment at different temperatures

177-178: The pre-treatment time also affects the density and diameter of CNTs on the catalyst. Consider: the catalyst pretreatment time also …

182: Figure 10. SEM images of formed – put the scale in

199-200: carbon atoms decomposed from precursor – carbon atoms freed from

200: came into contacting with metallic catalyst into contact

205: As soon as the gas is flown on the surface - As soon as the gas hits the surface

212-213: TEM imaging confirmed the presence of MWNTs with 6 walls

215: underlayer – support

220: investigated - investigation

225-226: both sentences have opposite meaning

Author Response

Thanks for your valuable and detailed comments. Please see the attachment for our response. 

Reviewer 3 Report

The authors have used a CVD method with iron foil as catalyst to produce multiwall carbon nanotubes (MWCNTs), but have not provided sufficient characterization of the samples produced. For example, the Raman spectra shown do not appear to be from MWCNTs. They have not assigned the lines observed and it is not clear how they determined from the Raman data that the MWCNTs produced are metallic. In addition, the SEM and TEM images presented are of poor quality. 

Author Response

Thanks for your valuable comments. We have revised the manuscript based on your suggestion. Please see the attachment for our response. 

Reviewer 4 Report

Minh Huynh et al. present an interesting article on catalyst pre-treatment effects on carbon nanotube synthesis from methane using thin stainless-steel foil as a catalyst by a chemical vapor deposition method. The topic itself is fascinating due to the relevance of these materials in our daily days. The present version required some attention before it accepted for publication.

Below, some aspects that need attention:

  1. Page 2, line 52: citing references is better and more professional
  2. Page 2, line 66: this reviewer would suggest the authors consider the uses of catalyst precursor instead of catalyst due to the nature of the process: the foil is pre-treat (change the surface properties) before entering contact the reactant. By IUPAC definition, we are talking here as a catalyst precursor. The real catalyst is present once the substrate, gas reactant, and catalyst precursor surface interact at the reaction conditions to promote product formation.
  3. Page 2, line 68: Why did not used the conventional method to get a bulk composition analysis?
  4. Page 3, line 85: section 2.3 should be before CNT synthesis
  5. Page 4, line 113-120: this section should be more descriptive about the Raman signal associate with common CNT. The information provided in Figure 3 is precious, and it is poorly described.
  6. Page 4, lines 123 & 128 are the same. Please rephrase accordingly.
  7. Page 5, line 141: catalyst precursor is more appropriate
  8. Page 6, line 154: catalyst surface is better
  9. Page 8, lines 184-187: a better description needs it. The authors lost the opportunity to illustrate the relevance of the Raman information on this work and how it supports the evidence of the uses of this methodology
  10. Page 9, line 194: TGA data, the graph should be included
  11. Page 9, line 197-206, references supporting information must be included

Author Response

(The authors gave the same response as above.)

Round 2

Reviewer 1 Report

Although authors of this paper carried out major revision on their manuscript, the overall results are still very poor. In my opinion, this quality of results and presentations cannot be accepted for publication in Nanomaterials. Therefore, I cannot recommend this paper for publication.

Reviewer 3 Report

The author's revised manuscript has the same issues regarding the Raman spectra and electron microscopy data  pointed out in my previous review.

Reviewer 4 Report

Thanks the authors, for addressing all comments presented in the previous review, This reviewer supports the publication of this manuscript

Good luck